# Interactive Segmentation by Considering First-Click Intentional Ambiguity

## ABSTRACT

Interactive segmentation task aims at taking into account the influence of user preferences on the basis of general semantic segmentation in order to obtain the specific target-of-interest. Given the fact that most of the related algorithms generate a single mask only, the robustness of which might be constrained due to the diversity of user intention in the early interaction stage, namely the vague selection of object part/whole object/adherent object, especially when there's only one click. To handle this, we propose a novel framework called **D**iversified **I**nteractive **S**egmentation **N**etwork (DISNet) in which we revisit the peculiarity of first click: given an input image, DISNet outputs multiple candidate masks under the guidance of first click only, it then utilizes a Dual-attentional Mask Correction (DAMC) module consisting of two branches: a) Masked attention based on click propagation. b) Mixed attention within first click, subsequent clicks and image *w.r.t.* position and feature space. Moreover, we design a new sampling strategy to generate GT masks with rich semantic relations. The comparison between DISNet and mainstream algorithms demonstrates the efficacy of our methods, which further exemplifies the decisive role of first click in the realm of interactive segmentation.

## CCS CONCEPTS

• **Computing methodologies → Image segmentation**.

## KEYWORDS

Interactive Segmentation, Multiple Output, First-click, Attention Mechanism

## 1 INTRODUCTION

Interactive segmentation can be viewed as a mutable and active instance segmentation task. Unlike general segmentation methods that predict solid mask for every latent object in an image, interaction-based method is capable of making single or multiple object selection in a human-in-the-loop manner, allowing users to provide prompts iteratively in order to focus on specific targets, plus eliminate the mislabeled regions such as holes and flaws, till harvesting satisfactory results. The advent of DIOS [45] has greatly promoted the relevant research process, numerous deep-learning based interactive segmentation algorithms have been proposed, covering various data categories (*e.g.*, natural image, medical image,

Permission to make digital or hard copies of all or part of this work for personal or classroom use is granted without fee provided that copies are not made or distributed for profit or commercial advantage and that copies bear this notice and the full citation on the first page. Copyrights for components of this work owned by others than the author(s) must be honored. Abstracting with credit is permitted. To copy otherwise, or republish, to post on servers or to redistribute to lists, requires prior specific permission and/or a fee. Request permissions from permissions@acm.org.
*ACM MM, 2024, Melbourne, Australia*
© 2024 Copyright held by the owner/author(s). Publication rights licensed to ACM.
ACM ISBN 978-x-xxxx-xxxx-x/YY/MM
https://doi.org/10.1145/nnnnnnn.nnnnnnn

infrared image) and interaction modalities (*e.g.*, click [7, 21, 43, 46], scribble [3], extreme points [34, 48], contour [1, 19, 36, 38], prompt [18, 44, 49]), which have also been popularized in data annotation, autonomous driving, as well as medical image analysis.

Over the last few years, researchers have made great efforts to enhance the classical interactive segmentation pipeline using prevailing computer-vision techniques. A large proportion of latest methods are embedded with attention mechanism (*e.g.*, transformer), which proves conducive for long-range context modeling. The pioneering work SimpleClick [27] adopts an MAE-pretrained ViT [14] consisting of 12~32 W-MSA layers [30]; SAM [18] and SEEM [49] also use ViT to pre-compute image features, while they further measure the cross-attention between image feature and encoded prompts in a two-way manner, thus facilitates mutual information flow.

It is also noteworthy that user ambiguity is another crucial aspect which has been gradually brought into focus. The meaning of ambiguity is that a single interaction may correspond to multiple system feedback, given that the annotated pixels by users might contain diversified semantics, which is ubiquitous in object/set of objects that is commonly characterized by structural hierarchy or spatial adjacency, such as desk *vs.* book on the desk or man *vs.* camera in man's hand. In terms of interaction modality, the mainstream click-based methods become susceptible to vague selection of mask prototypes compared with scribble-based or contour-based counterparts, since click possesses the sparsest prior knowledge that it is hard to indicate the accurate range of a specific object. Previous methods such as LD [21] and MultiSeg [22] attempts to tackle this issue using additional output channels, while recent study PiClick [46] reformulates this task on the basis of DETR [2], using solid number of object queries to generate mask proposals, which greatly surpasses the former. However, its weakness is also obvious: a) The rapid decline of ambiguity in the early rounds leads to output convergence (diverse to single), which causes the waste of object queries and its highly time-consuming computation (mostly $O(n^2)$). b) The generation of diverse GT masks is based on random merging strategy, hence quite a few semantic or spatial correlation between those masks, which hinders performance.

Our work makes further investigation into how to elaborately design an architecture where diversified user intentions are sufficiently parsed during interactive segmentation. We revisit and summarize two peculiar features of first click: a) Maximum ambiguity, means the number of diverse mask predictions reaches a peak when there's only one click. b) Contextual continuity, means the role of subsequent clicks is to refine details based on selected mask proposal (serves as initial template). We propose Diversified Interactive Segmentation Network (DISNet), a novel framework that decouples the classical segmentation procedure into two sections—proposal network and refinement network (see Figure 1). The main body of proposal network is Mask2Former [5], in which

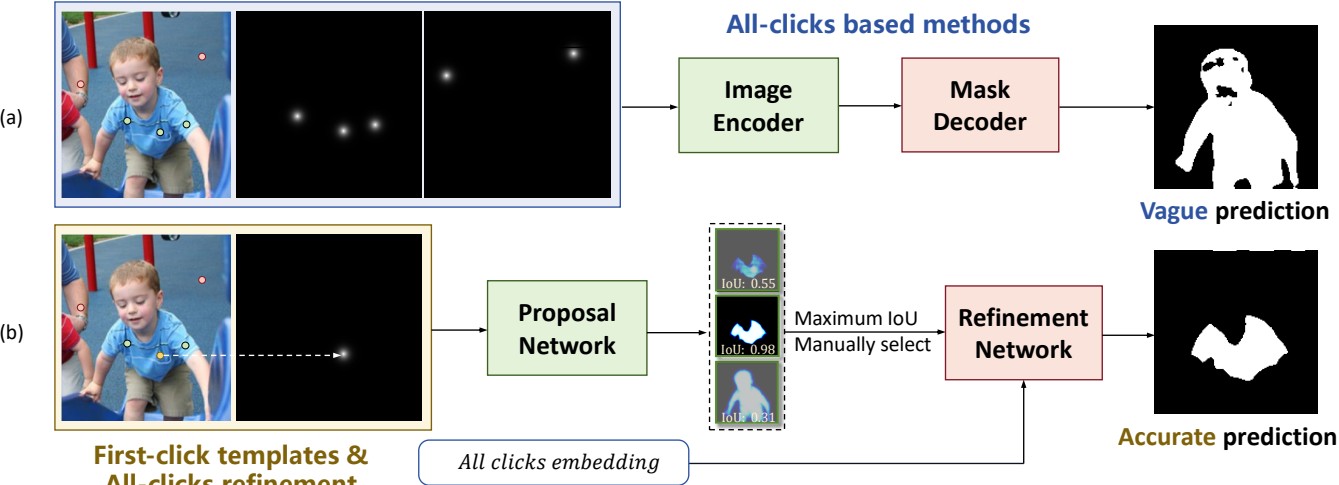

**Figure 1: (Top row) Typical interactive segmentation doesn't account for latent diversity in user intentions, causing flaws and holes in prediction mask. (Bottom row) In our method, first click takes the role of generating all plausible proposals, a coarse mask is selected as a guiding template based on IoU or manually by user, which is then refined by subsequent clicks to get final accurate result.**

we use RGB image combined with 1-channel first click map as input to get diverse mask predictions. Later, a single mask is picked out according to the largest IoU between predictions and GT mask, then the selected mask/mask token, image feature, together with encoded full clicks are sent to refinement network, where we introduce Dual-attentional Mask Correction (DAMC) which can be regarded as a variant of two-way transformer in SAM, including a masked click-attention module and a first-click guidance module. The former updates mask token using cross-attention between token and image feature, where the affinity matrix is reweighted by endowing its elements in the proximity of clicks with a larger value to emphasize click propagation. The latter updates image feature by computing a relational vector between first click and other clicks, then utilizes it to perform channel-wise activation with image feature. Both modules aim to strengthen cross-over information flow in terms of robustness and efficiency. Finally, we design a novel principle to generate sequence of diverse ground-truth masks on SBD [13] and LVIS [11] datasets. Evaluation on six benchmarks shows outstanding performance compared with the existing methods. We achieve 3.07 NoC%85 with 5.11 NoC%90 when trained with LVIS, which outperforms PiClick [46] (3.11 NoC%85 with 5.32 NoC%90), which is the current SOTA method.

We summarize our contributions as follows:

- We introduce an interactive segmentation framework DIS-NET that features multiple-output and first-click design, in which we fully exploit the properties of user's first click: a) to represent latent, diverse user intentions (maximum ambiguity). b) to guide and constrain the impact of successive clicks (contextual continuity).
- We propose Dual-attentional Mask Correction (DAMC) component, a modified two-way transformer used to mutually measure the attention among image features, click features and selected token/mask proposal, which is proved to be

capable of manifesting the decisive effect of user's first click in terms of information flow.
- We propose a novel mask-sampling method in order to match our new framework. During training, we generate a set of semantic-correlated ground-truth mask proposals given the position of first click, which are then used to supervise the output from the first stage of DISNET.
- Extensive comparisons with former works, visualization and ablation studies have demonstrated the necessity of our network design from macro to micro level. We conduct these experiments on six datasets using two evaluation metrics.

## 2 RELATED WORK

### 2.1 Interactive Segmentation

Interactive segmentation (IS) takes account of the human guidance to provide single, class-agnostic instance mask, which has been a long-standing topic since the advent of Intelligent Scissor [36] in 1995. During that period, researchers mainly focus on energy optimization methods, *e.g.*, GrabCut [1] and random walk [10]. Those methods leverage low-level feature only, which is sub-optimal when confronting complex scenes. In 2016, the first deep-learning based algorithm DIOS [45] makes a remarkable breakthrough in IS. In this work, positive and negative clicks are encoded into two-channel distance maps concatenated with input image, an arbitrary segmentation model (*e.g.*, FCN [31]) takes it as input to get final result in an end-to-end manner. Later, ITIS [33] proposes iterative training strategy to simulate real-word interaction process in training phase, which is then revisited and further modified by [41]. In terms of interaction type, DEXTR [34] uses extreme points in four directions to indicate a compact range of object compared to click, while in IOG [48], a bounding box is drawn for coarse localization, then clicks are added inside the box to obtain fine-grained mask. In recent studies,

Figure 2: An overview of our proposed DISNet. The whole network is a combination of proposal generation network $P$ and mask refinement network $R$. At first, multiple predictions based on first click are provided by $P$ to represent diverse intentions. Next, a mask is selected based on user preference or the largest IoU score with GT, which will be further refined in $R$ using whole clicks.

attention-based design has gradually become mainstream. CDNet [4] disseminates pixel features located at positive/negative clicks to other pixels using self-attention. iFPN [47] adopts sparse GNNs to propagate click information in a long-range manner. SimpleClick [27] and iSegFormer [28] are the pioneering works to combine transformer with IS, which greatly motivate relevant research such as iCMFormer [20] and InterFormer [16].

## 2.2 Interactive Segmentation with Diverse Output

Originated from LD [21], researchers attempt to combine IS with multiple choice learning[12], which is a strategy that guides the model to generate more than one feasible solutions and select from one of these. In LD, the number of output channels is altered to 6, meanwhile only the minimum loss among those channels is back-propagated so as to learn discrepancy within channels. MultiSeg [22] provides multiple scale-aware masks by computing loss in anchor-truncated areas. SAM [18] utilizes a specific prompt token as a signal to judge whether to return diverse or single mask based on the level of ambiguity (number of click), PiClick [46] enables 7 object queries to learn output diversity under the supervision of multiple GT masks. In comparison, our method generates proposals only when user clicks for the first time. Even though clicks up to 2~3 rounds may still contain intentional uncertainty, we argue that a trade-off between low-time cost and performance uprising is of great necessity in network design.

## 2.3 Interactive Segmentation with First Click

IS methods concerning about the utility of first click are quite few till now. In FCANet [26], first click is regarded as a coarse prior for subsequent clicks to refine, in which they supervise first-click-only prediction mask as a subtask using auxilliary loss. EMC-Click [7] improves FCANet by proposing two novel correction modules which boosts performance. The uniqueness of first click is still lack of sufficient exploration and it is meant to bring about a brand-new perspective when diving into relevant research.

# 3 METHODOLOGY

## 3.1 Preliminary and Overview

Given an input image $I \in R^{H \times W \times 3}$ with user-annotated set of pixels $C = \{(u_i, v_i, p_i) | i = 1, 2, \cdots, c\}$, where $(u_i, v_i) \in [0, W] \times [0, H]$ and $p_i \in \{0, 1\}$ denotes the coordinates and property (i.e., $p_i = 1$ for positive and $p_i = 0$ for negative) of the $i^{th}$ click, an encoding function $Enc2D(\cdot)$ converts those clicks into a 2D pattern, e.g., distance map (denoted as $S \in R^{H \times W \times 2}$), which is concatenated with image to form a 5-channels input. In a standard segmentation network $f$, a pretrained image encoder scales down the input resolution by using stride-2 convolutions or pooling (e.g., ResNet-50), followed by stacks of conv layers to extract high-level feature, which is then upsampled by a decoder structure for semantic comprehension. The final output is a sigmoidized mask $M \in [0, 1]^{H \times W}$ which indicates the fine-grained location of user-interest object. Later, user is allowed to add more clicks targeted on the mislabeled region, causing the iterative change of prediction together with input clicks. Therefore, we formulate the classical interactive segmentation pipeline as follows:

$$M^t = f\left(I, S^t, M^{t-1}; \theta_f\right) \tag{1}$$

where $t \in \{1, 2, \cdots, T\}$ denotes the $t^{th}$ interaction round, $\theta_f$ denotes the network parameter.

However, Formula 1 is insufficient when conditioned on multiple predictions combined with the prominence of first click, which characterizes the main architecture of DISNet. As is shown in Figure 2, the whole network is split into two stages for proposal generation and mask refinement, respectively. In the first stage (denoted as proposal network $P$), click in the first round (i.e., $t = 1$) is distinguished from other clicks in $C^t$ due to the diverse user intentions it brings about, denoted as $C_F$, which is then converted by $Enc2D(\cdot)$ to form a single-channel disk map $S_F$ (Here we suppose that $C_F$ is always positive so we omit the second channel for negative clicks, since user is fond of clicking around the center of object at first, according to [26]). Similarly, we take the image $I$ with $S_F$ as input, deliver to the image encoder, while a Mask2Former decoder is used to generate $N$ diverse, ambiguity-aware mask proposals

$\mathcal{M} \in [0, 1]^{H \times W \times N}$ using trainable object queries $Q \in R^{N \times d}$, together with a pixel decoder which produces multi-scale features on the basis of encoder output. Finally, a specific mask $\mathcal{M}_s$ is selected from $\mathcal{M}$ as input to stage 2 (denoted as refinement network $R$) to receive further correction. The above procedure can be formulated as follows:

$$\mathcal{M}_s = \phi_s\left(\mathcal{M}\right), \mathcal{M} = P\left(\mathcal{I}, \mathcal{S}_F; \theta_p\right) \tag{2}$$

where $\phi_s\left(\cdot\right)$ denotes mask selection principle (see Section 3.2), note that $Q$ is part of $\theta_p$ as network parameters hence absence in the variable list. It is obvious that $\mathcal{M}_s$ remains solid after the first click is given, which won't be altered *w.r.t.* $C^t$ or $\mathcal{M}^{t-1}$. Therefore, DISNet only needs to focus on mask refinement from the second round, which enormously reduces the cost of dense matrix operation in stage 1.

Next, subsequent click is added in order to refine the erroneous part of $\mathcal{M}_s$. The object query $Q_s \in R^d$ (takes charge of predicting $\mathcal{M}_s$), together with 4x resolution feature map produced by pixel decoder (denoted as $\mathcal{F}^{-1} \in R^{H/4 \times W/4 \times d}$), are reused in stage 2 for consistency. Specifically, two elaborately designed attention module followed by respective type of FFNs act as a modified mask decoder in SAM to output the final mask $\mathcal{M}_f$, which is formulated as below:

$$\mathcal{M}_f^t = R\left(Q_s, \mathcal{F}^{-1}, \mathcal{M}_s, \mathcal{S}^t, \mathcal{P}^t; \theta_R\right) \tag{3}$$

where $\mathcal{P}^t \in R^{C \times d}$ is generated using $Enc1D(\cdot)$ function to get a linear representation of $C^t$. We'll clarify this function and the detailed mechanism of stage 2 in Section 3.3.

## 3.2 Proposal Generation

Masks with all possible semantic combinations (user intentions) will be provided in this stage. We mainly follow the design of PiClick [46] except for some minor adjustments.

**Image encoder**. A general encoder is often used to extract image feature as a preprocess in almost all computer vision task. In our method, a plain ViT pretrained with Masked Image Modeling (MIM) [14] is adopted, which includes a patch embedding layer and several window-based multi-head self-attention layers (W-MSA). We obtain feature with 16x resolution and 784 channels (ViT-B version) from the encoder.

**Mask decoder**. We constitute our mask decoder with a pixel decoder to get multi-level features $\{\mathcal{F}_i | i = 1, 2, 3\}$, plus several Mask2Former [5] decoder layers which involves: a) self-attention within object queries $Q$. b) cross-attention between $Q$ and $\mathcal{F}_i$. c) feedforward network (FFN). Predictions are obtained by matrix calculating (convolving) 4x feature $\mathcal{F}^{-1}$ with the updated $Q$, while a bipartite matching loss is measured between prediction masks and relation-aware GT masks (see Section 3.4). The specific mask to refine in stage 2 is picked out according to its largest IoU with all GT masks(during training and evaluation) or user selection (during inference). Note that we abandon the design of parallel IoU predictor (Target Reasoning Module) in PiClick [46] since it is an ill-posed problem to produce accurate IoU without external guidance under first-click-only circumstance.

## 3.3 Dual-attentional Mask Correction

Motivated by SAM [18], we propose masked click attention module (regarded as modified image-to-token) with first-click guidance module (regarded as modified token-to-image) to progressively renovate image features and token, as illustrated in Figure 3.

**Masked click attention**. We start by introducing the masked attention scheme adopted in DETR series[5, 9]. Suppose we have $Q$, $K$, $V$ that satisfies $Q = \varphi(Q_s)$, $K = \psi(\mathcal{F}^{-1})$, $V = \Theta(\mathcal{F}^{-1})$, where $\varphi$, $\psi$ and $\Theta$ are linear transformations. In masked attention, the value of $QK^T$ is summed with a modulation term $M$ before softmax operation.

$$\mathcal{Z}_{MA} = \text{softmax}\left(\frac{QK^T}{\sqrt{d}} + M\right)V \tag{4}$$

The above $M$ enforces all pixels in the background (sigmoid value less than 0.5) to be infinitesimal so that information flow is limited into foreground pixels only. When combined with our work, $M$ is replaced by $\log(\mathcal{M}_s)$, where $\mathcal{M}_s$ is the selected mask proposal based on first click. There're mainly two concerns: a) $\mathcal{M}_s$ serves as an initial template to provide coarse localization, which accelerates the convergence speed of training. b) $\mathcal{M}_s$ is solid, means the contextual continuity of first click could be kept. We use $\log(\cdot)$ instead of binary threshold to ensure smooth transition, which could also contribute to efficient training.

We also notice that positive/negative click is also the subset of foreground and background pixels. From the click perspective, mask is somewhat a zone of propagation or diffusive growth start from seeds (clicks). Therefore, another modulation term is required to manifest the saliency of clicks in masked attention. Hence, we generate two linear decay maps (denoted as $\mathcal{G}_p$ and $\mathcal{G}_n$) according to click positions, which satisfies:

$$\mathcal{G}_{p/n}(i, j) = 1 - \frac{\min\left(r_0, \phi\left(C_{p/n}, p_{ij}\right)\right)}{r_0} \tag{5}$$

where $C_{p/n}$ is the positive/negative click set, $\phi(\cdot)$ is euclidean distance, $r_0$ is a radius to control the rate of decay (We set 5 for other clicks and 15 for the first click). Lastly, the masked click attention (MCA) is formulated as below:

$$\mathcal{Z}_{MCA} = \mathcal{Z}_p + \mathcal{Z}_n \tag{6}$$

$$\mathcal{Z}_p = \text{softmax}\left(\frac{QK^T}{\sqrt{d}} \odot \mathcal{G}_p + \log\left(\mathcal{M}_s\right)\right)V \tag{7}$$

$$\mathcal{Z}_n = \text{softmax}\left(\frac{QK^T}{\sqrt{d}} \odot \mathcal{G}_n + \log\left(1 - \mathcal{M}_s\right)\right)V \tag{8}$$

where $\odot$ means Hadamard product. It is evident that $\mathcal{G}_{p/n}$ is utilized to reweight $QK^T$ under the constraint of $\mathcal{M}_s$, thus the feature of pixels nearby those clicks are the most likely to be gathered, which realizes delicate, oriented information flow from image to token.

**First-click guidance**. We strive to forge a novel token-to-image module that satisfies: a) low computation cost. b) a thorough exploitation of the first click. Reviving Formula 3, clicks are encoded into 1D vectors $\mathcal{P}^t$ by initializing two trainable indicators representing the positive/negative property, a positional embedding function is used to transform clicks coordinates into a dense vector form, which is then added with indicator. Followed by concatenation and

Figure 3: Detailed mechanism of Masked click attention (MCA) and First-click guidance (FCG) module in DAMC.

linear projection, one could obtain click-augmented features which we denote as follows:

$$\mathcal{F}' = \text{LinearProj}\left(\mathcal{F}^{-1} \oplus \mathcal{P}^t\right) \quad (9)$$

A latent issue is that clicks may be sub-optimal such as clicking in regions with blurred patches or low lighting, even the click itself could be erroneous due to user's negligence, i.e., mark negative on foreground. A non-trivial solution is to exclude those clicks or weaken their influence to the network. Naturally, the first click contains the user's most primary impression of object's global structure, which is qualified for guidance prior. Therefore, we measure the feature-space correlation between the first click and other clicks (denoted as relational vector $\mathcal{V}$) in order to pose a constraint that only the click which shares similarity with the first click could contribute to mask output. To this end, we split $\mathcal{F}'$ into $\mathcal{F}'_{fc}$, $\mathcal{F}'_{pos}$ and $\mathcal{F}'_{neg}$, then cross-attention within the three is computed as follows, where MHA means multi-head attention:

$$\mathcal{V}_{pos/neg} = \text{MHA}\left(\mathcal{F}'_{fc}, \mathcal{F}'_{pos/neg}\right) \quad (10)$$

Later, we integrate $\mathcal{V}$ with image feature $\mathcal{F}^{-1}$ by utilizing SENet [15], which consists of two fully-connected layers, a sigmoid function, and a channel-wise multiplication between $\mathcal{V}$ and $\mathcal{F}^{-1}$. This is fairly efficient compared with the original token-to-image module in SAM, in which a spatially cross-attention between $\mathcal{F}^{-1}$ and all clicks embedding is implemented. Similar to MCA, we use $\mathcal{M}_s$ as a second constraint to limit the scope of $\mathcal{V}$. Thus, the first-click guidance (FCG) module is formulated as below:

$$\mathcal{Z}_{FCG} = \mathcal{Z}_p + \mathcal{Z}_n \quad (11)$$

$$\mathcal{Z}_p = \text{SENet}\left(\mathcal{F}^{-1}, \mathcal{V}_{pos}\right) \odot \mathcal{M}_s \quad (12)$$

$$\mathcal{Z}_n = \text{SENet}\left(\mathcal{F}^{-1}, \mathcal{V}_{neg}\right) \odot \left(1 - \mathcal{M}_s\right) \quad (13)$$

Finally, we adopt a depth-wise ConvFFN module to make up for the lack of spatial attention, which consists of two $1 \times 1$ conv layer and a $3 \times 3$ depth-wise separable conv layer. This is a common practice in many real-time vision transformers.

### 3.4 Relation-aware Training Samples

The generation of diverse ground-truth masks is a non-trivial task for the supervision of proposal network in DISNet. Based on SBD [13] and LVIS [11] datasets, we attempt to discover the best strategy to measure the possible spatial/semantic correlation within all masks in an image, which we categorize into proximity-based method and hierarchy-based method. A concrete algorithm description of mask sampling with clicks using the two methods is shown in supplementary material.

**Hierarchy-based method**. Thanks to the hierarchy tree provided in LVIS that enable us to construct a sequence of object masks with rich semantic relations. In this scenario, a scene could be decoupled into different levels of scene nodes (may contain multiple objects for the root, and object part for the leaf) to insinuate the possible subject-predicate-object (SPO) relations among them. Concretely speaking, we adopt a bottom-up strategy where we first randomly pick out a center object (e.g., an apple) and locate its node level. Then we traverse all of its parent nodes (e.g., a man with an apple) using depth ordering, until the level of node reaches the top (e.g., a man with an apple is sitting on a chair). Finally, the center object along with parents are added into an empty list of diverse ground-truth masks in an inner-outer manner. We sample the first click based on the subtracted region between the center object mask and all of its children masks, while a random mask is selected to supervise the refinement network.

**Proximity-based method**. Mask sampling using hierarchy tree is not feasible in SBD. Therefore, we simply judge whether two arbitrary masks are spatially proximate by calculating the overlapped pixels after a $3 \times 3$ mask dilation, which is not guaranteed for semantic relations. We use all combinations of 1-hop neighbors to get the list of diverse samples.

## 4 EXPERIMENTS

### 4.1 Datasets and evaluation metrics

We conduct our experiments on the following datasets: GrabCut [39], Berkeley [35], DAVIS [37], SBD [13], PascalVOC [8] and a combination of COCO [24] and LVIS [11]. The GrabCut dataset contains 50 images with single object. The Berkeley dataset consists

**Table 1: A comprehensive comparison between the mainstream algorithms and our method. Methods marked with ↑ means the first-click is treated specially during segmentation, while ↓ means methods with multiple outputs. The best result is marked with blue (trained with SBD) and red (trained with COCO+LVIS). It is obvious that our method possesses both of the two characteristics.**

| Method | Dataset | Backbone | GrabCut | | Berkeley | | DAVIS | | SBD | |
|---|---|---|---|---|---|---|---|---|---|---|
| | | | NoC%85 | NoC%90 | NoC%85 | NoC%90 | NoC%85 | NoC%90 | NoC%85 | NoC%90 |
| DOS w/o GC[45] | VOC | FCN-8s | 8.02 | 12.59 | - | - | 12.52 | 17.11 | 14.3 | 16.79 |
| DOS with GC[45] | VOC | FCN-8s | 5.08 | 6.08 | - | - | 9.03 | 12.58 | 9.22 | 12.80 |
| DEXTR[34] | VOC | ResNet-101 | - | - | - | - | - | - | - | - |
| FCANet[26] ↑ | VOC | ResNet-101 | - | 2.14 | - | 4.19 | - | 7.90 | - | - |
| MultiSeg[22] ↓ | VOC | ResNet-101 | - | 2.30 | - | 4.00 | - | - | - | - |
| IOG[48] | VOC | ResNet-50 | - | - | - | - | - | - | - | - |
| f-BRS-B[40] | SBD | ResNet-101 | 2.30 | 2.72 | - | 4.57 | 5.04 | 7.41 | 4.81 | 7.73 |
| LD[21] ↓ | SBD | VGG-19 | 3.20 | 4.79 | - | - | 5.59 | 9.57 | 7.41 | - |
| IA+SA[42] | SBD | ResNet-101 | - | 3.07 | - | 4.94 | 5.16 | - | - | - |
| FocusCut[25] | SBD | ResNet-101 | 1.46 | 1.64 | - | 3.01 | 3.40 | 5.31 | 4.85 | 6.22 |
| CDNet[4] ↓ | SBD | ResNet-101 | 2.42 | 2.76 | 1.47 | 2.06 | 5.33 | 6.97 | 4.73 | 7.66 |
| RITM[41] | SBD | HRNet-18s | 1.76 | 2.04 | 1.87 | 3.22 | 4.94 | 6.71 | 3.39 | 5.43 |
| FCFI[43] | SBD | ResNet-101 | 1.64 | 1.80 | - | 2.84 | 4.75 | 6.48 | 3.26 | 5.35 |
| SimpleClick[27] | SBD | ViT-B | 1.58 | 1.66 | 1.55 | 2.37 | 4.10 | 5.48 | 3.24 | 5.43 |
| Ours ↑↓ | SBD | ViT-B | 1.54 | 1.68 | 1.39 | 2.07 | 4.07 | 5.26 | 3.39 | 5.24 |
| RITM[41] | C+L | HRNet-32 | 1.46 | 1.56 | 1.43 | 2.10 | 4.11 | 5.34 | 3.95 | 5.71 |
| EMC-Click[7] ↑ | C+L | SegF-B3 | 1.42 | 1.48 | - | 2.35 | 4.49 | 5.69 | 3.44 | 5.57 |
| FCFI[43] | C+L | HRNet-18s | 1.50 | 1.56 | - | 2.05 | 3.88 | 6.24 | 3.70 | 5.16 |
| SimpleClick[27] | C+L | ViT-B | 1.38 | 1.48 | 1.36 | 1.97 | 3.66 | 5.06 | 3.43 | 5.62 |
| ICMFormer[20] | C+L | ViT-B | 1.42 | 1.52 | 1.40 | 1.86 | 3.40 | 5.06 | 3.29 | 5.30 |
| InterFormer[16] | C+L | ViT-B | - | 1.48 | - | 1.97 | - | 5.06 | 3.43 | 5.62 |
| PiClick[46] ↓ | C+L | ViT-B | 1.18 | 1.24 | 1.17 | 1.78 | 3.42 | 4.60 | 3.11 | 5.32 |
| Ours ↑↓ | C+L | ViT-B | 1.16 | 1.16 | 1.22 | 1.75 | 3.80 | 4.51 | 3.07 | 5.11 |

of 96 images with 100 instances. The SBD dataset is divided into 8498 samples for training, and 2857 for validation. The DAVIS dataset contains 50 videos primarily designed for video-based segmentation task, here we follow [7, 27, 41] to randomly sample 345 frames for testing. The PascalVOC dataset contains 1449 testing images with 3427 instances. The COCO+LVIS is a compound dataset which consists of 118K images with high-quality annotations of about 1.2M instances. In our work, the SBD and COCO+LVIS are used for training purposes while the rest is for testing.

We follow the commonly used evaluation protocol for interactive segmentation, including a) Mean intersection-over-union (mIoU), which measures the percentage of pixels in the overlapped regions of predicted mask and ground truth. b) Number of clicks (NoC), which measures the least number of clicks required to reach a given IoU threshold $x$ (denoted as NoC%$x$, where $x$ takes the value of 85 or 90 as a common practice in previous work [4, 20, 29, 48]).

## 4.2 Implementation details

Following [46], we adopt a ViT-B as backbone (pretrained by MAE) together with 3 Mask2Former decoder layers for proposal generation. The selection of proposal is based on the largest IoU with current ground-truth mask, or manually by user preference when the model is online for real world use. In the latter part of DISNet,

the number of DAMC module is commonly set to 2 to prevent latent computation cost. During the training process, We supervise the whole network with a bipartite matching loss [2] for stage 1 and a normalized focal loss (NFL) [23] for stage 2, while the learning rate $lr$ is set to $5e^{-5}$ and $5e^{-6}$ for two stages, respectively. We adopt Adam [17] optimizer with a momentum of $\beta_1 = 0.9$ and $\beta_2 = 0.99$, followed by a cosine-annealing scheduler [32] for progressive $lr$ decay (we set the warmup step to 2 epochs, and initial $lr$ to $5e^{-8}$). We train 55 epochs on SBD dataset and 70 epochs on COCO+LVIS dataset, both using a batch size of 32. For a training sample, we randomly crop and resize it to a resolution of $448 \times 448$, followed by classical data augmentations, *i.e.*, random brightness/contrast, horizontal/vertical flipping, *etc*. All the experiments are conducted on Ubuntu-18.04 platform with 4 RTX 4090 GPUs, while our main code is constructed on two open-source projects (*i.e.*, MMSegmentation [6] and RITM [41]).

During inference when the first click is positioned, we truncate the path to refinement network $R$ so that the selected mask (output from proposal network $P$) is chosen for metric evaluation, since we notice a subtle performance drop around 0.3 NoC if we evaluate the refined mask from network $R$ instead. Clearly, only the first click itself provides limited prior knowledge to fix local details. Starting from second click, we reuse the output from network $P$ such that

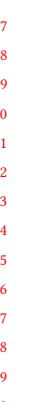
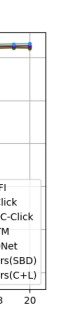
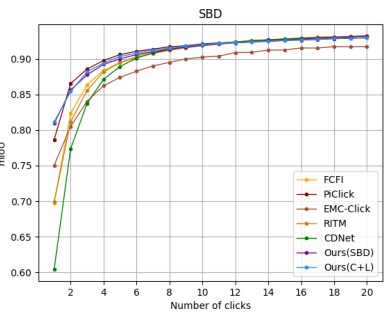
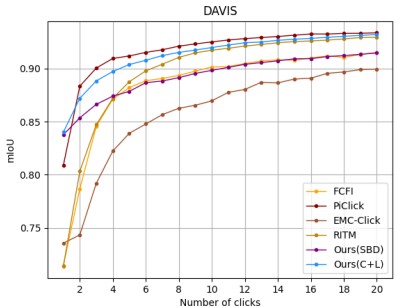

**Figure 4: Curves to measure the change of mIoU with the number of clicks on 3 datasets. It is manifest that our method outperforms the others especially when there's only first click, which proves the efficacy of first click design.**

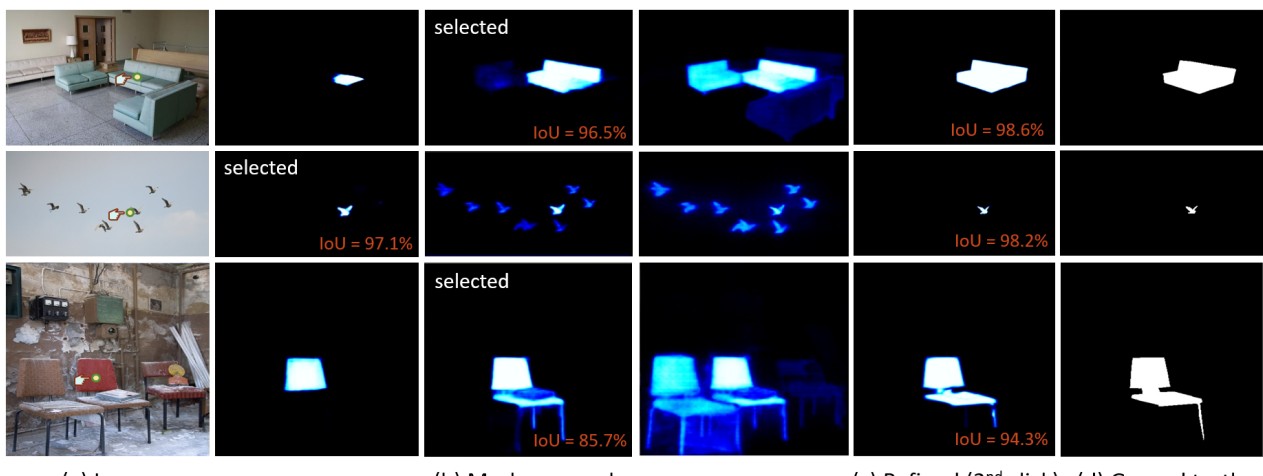

(a) Image  (b) Mask proposals  (c) Refined (2nd click)  (d) Ground truth

**Figure 5: Visualization of output at each stage. The position of first click is marked with yellow dot.**

only $R$ is utilized, which avoids redundant computation cost. This time-efficient strategy is similar to [7].

## 4.3 Comparisons with State-of-the-arts

We make a thorough comparison with the mainstream IS algorithms, the Number of Click (NoC) result is shown in Table 1. To be fair, we don't emphasize on the effect of a stronger backbone or a strong dataset so we group methods with the above similar setting for convenient analysis. We notice that our work outperforms most of the previous works, where there's a slight improvement of 0.02 to 0.08 compared with PiClick on GrabCut and a huge improvement over EMC-Click [7] (about 0.2 to 0.3)—the latter method also attempts to measure the peculiarity of first click but failed to sufficiently leverage this information. When in terms of dataset, we argue that a high quality masks annotation is indispensable since we surpass our own counterpart (trained by SBD) at an astonishing value of 0.4 to 0.5. Reviving the hierarchy mask sampling strategy in COCO+LVIS dataset, it is likely to make great contribution not only for the proposal stage, but also a guidance prior to the final output. In terms of multiple output mask design, CDNet [4] utilizes

an auxiliary branch to learn the correlated semantic, yet compared with PiClick [46] or our method, it fails to converge well enough due to the lack of diverse ground truth masks.

Curves measuring the change of mIoU with respect to number of clicks are shown in Figure 4. We make a brief comparison among up to 7 methods on GrabCut, Berkeley, SBD and DAVIS datasets, respectively. It is evident that our method performs the best especially at first click.

We visualize the outputs from different stage in our framework, as is illustrated in Figure 5. A specific proposal is selected based on largest IoU with ground truth, then further refined by successive clicks. We notice that it is capable of harvesting result with sufficient accuracy starting from the second click.

We also strive to analyze two main properties in our work, *i.e.* first-click and ambiguity-aware mask proposals. As shown in 6, we visualize and compare the proposals (PiClick) or final result (FCANet, EMC-Click) at first click, together with IoU with respect to GT (red region in the middle). Our method tends to produce more accurate proposals compared to others (*i.e.*, we segment the click position into sofa cushion/sofa/adjacent sofas).

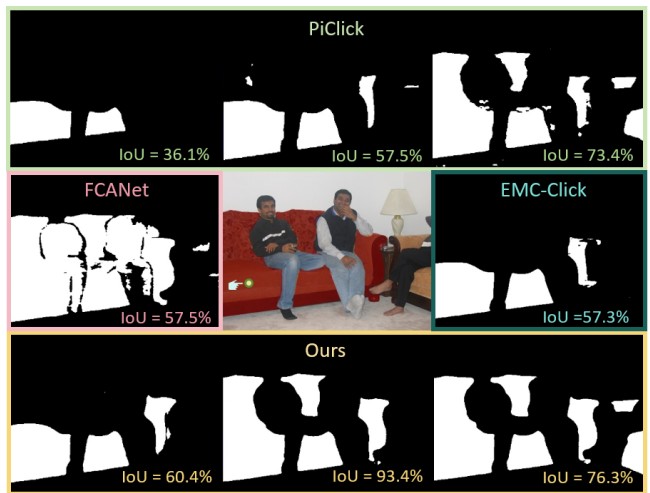

**Figure 6: Comparison of mask proposals or final mask when a first click is given. In contrast to other methods, our work is likely to produce high quality proposals with a larger IoU.**

## 4.4 Ablation Study

In this chapter, we conduct several ablation studies using Berkeley and DAVIS dataset, in which we take a deep view over each component of DISNet.

**Impact of whole network.** We start by progressively adding core components to a plain baseline (Here we choose SAM [18]), including FC (*i.e.*, first-click) and MO (*i.e.*, multiple output) design in proposal network $P$, together with MCA and FCG module in refinement network $R$. In addition to NoC, we utilize a novel metric called *Number of Failure* (NoF%$x$) which measures the number of cases where the segmentation mask could not reach a given mIoU $x$% within the maximally allowed number of clicks (default is 20). Reviving result in Table 2, we could notice that model with FC achieves a major breakthrough in NoC, which indicates the importance of first click. Furthermore, a proper integration of these four components leads to the best result, which means they're tightly organized with less redundancy during the whole pipeline.

**Impact of MCA.** Targeted on the rate of decay around each click, we design a scheme to record the effect of MCA under different combination of radius value $r_0$ for first/other clicks (shown in Table 3). We conclude that setting this two value to 15 and 5 could yield the best result.

**Impact of FCG.** To measure the utility of FCG. We split it into two components: a) whether to use click-augmented feature (simplified as AC) instead of image feature. b) whether to use ConvFFN module in its following. The result in Table 5 demonstrates that each component is indispensable to the final contribution of evaluation metrics.

**Impact of mask sampling.** We apply SBD and COCO+LVIS training dataset with two relation-aware sampling methods, while we record our result in Table 4. Evidently speaking, hierarchy-based method is more likely to produce accurate, semantic-exclusive diverse predictions than proximity-based method, which has been also demonstrated in Table 6.

**Table 2: Analysis of First-Click-Guidance (FCG) module on GrabCut and Berkeley dataset. *AC* means click-augmented feature.**

| FCG components | | GrabCut | | Berkeley | |
|---|---|---|---|---|---|
| AC | ConvFFN | NoC%85 | NoC%90 | NoC%85 | NoC%90 |
| | | 1.31 | 1.35 | 1.29 | 1.84 |
| ✓ | | 1.17 | 1.18 | 1.23 | 1.79 |
| | ✓ | 1.17 | 1.21 | 1.25 | 1.81 |
| ✓ | ✓ | 1.16 | 1.16 | 1.22 | 1.75 |

**Table 3: Analysis of Mask-Click-Attention (MCA) module on Berkeley and SBD dataset. We evaluate on different combination of radius $r_0$ for first/other click, respectively.**

| MCA radius $r_0$ | | Berkeley | | SBD | |
|---|---|---|---|---|---|
| First click | Other click | NoC%85 | NoC%90 | NoC%85 | NoC%90 |
| 5 | 5 | 1.51 | 2.07 | 3.44 | 5.25 |
| 15 | 5 | 1.39 | 2.07 | 3.39 | 5.24 |
| 15 | 15 | 1.39 | 2.09 | 3.39 | 5.25 |

**Table 4: Comparison with respect to relation-aware sampling strategy. Due to the lack of hierarchy tree for SBD, we only implement proximity-based method on this dataset.**

| Dataset & Sampling | SBD | | | DAVIS | | |
|---|---|---|---|---|---|---|
| | NoC%85 | NoC%90 | NoF%90 | NoC%85 | NoC%90 | NoF%90 |
| SBD & Proximity | 3.39 | 5.24 | 112 | 4.07 | 5.26 | 61 |
| C+L & Proximity | 3.11 | 5.23 | 109 | 3.95 | 4.77 | 49 |
| C+L & Hierarchy | 3.07 | 5.12 | 107 | 3.80 | 4.51 | 45 |

**Table 5: A thorough plug-in analysis for each of the core component in our work. *FC/MO* means first-click/multiple output design.**

| Component | | | | Berkeley | | SBD | | DAVIS | |
|---|---|---|---|---|---|---|---|---|---|
| FC | MO | MCA | FCG | NoC%90 | NoF%90 | NoC%90 | NoF%90 | NoC%90 | NoF%90 |
| | | | | 2.35 | 9 | 9.76 | 216 | 7.52 | 168 |
| ✓ | | | | 2.09 | 9 | 7.88 | 189 | 7.32 | 146 |
| ✓ | ✓ | | | 2.09 | 7 | 7.53 | 177 | 6.41 | 93 |
| ✓ | ✓ | ✓ | | 1.87 | 5 | 5.98 | 145 | 5.25 | 59 |
| ✓ | ✓ | | ✓ | 1.89 | 6 | 5.35 | 133 | 5.33 | 63 |
| ✓ | ✓ | ✓ | ✓ | 1.75 | 2 | 5.11 | 107 | 4.51 | 45 |

## 5 CONCLUSION

In this paper, we propose a two-stage interactive segmentation method DISNet, where the peculiarity of first interaction click is highlighted as maximum intentional ambiguity, together with contextual continuity. A novel refinement network DAMC further corrects details of the selected mask from the proposal network, which proves robust and efficient as well. Moreover, our proposed diverse ground-truth sampling strategy plays a crucial role in real-world simulation of user intentions. Extensive comparison and ablation studies demonstrates state-of-the-art performance on several datasets, which further exemplifies the decisive role of first click in the realm of interactive segmentation.

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
