# OpenReview forum: "Interactive Segmentation by Considering First-Click Intentional Ambiguity"
_acmmm.org/ACMMM/2024/Conference — MM2024 Poster_

### Official Review · Reviewer_uD9Y · 2024-05-24

**Rating:** 2
**Confidence:** 4

**Summary:**

To solve the ambiguity caused by the first click in the task of interactive image segmentation, the paper propose a novel framework called DISNet to better handling the peculiarity of  the first click.

**Strengths:**

1.The performance seems good.
2.This is the first work to treat the first click specially and output multiple result.

**Limitations:**

The biggest issue is that the results obtained after the first click in this paper were selected using the IoU metric, which assumes that the user has already obtained GT, which is not consistent with real-world scenarios.
Therefore, the performance data in this paper are unreliable. For example, the NoC of GrabCut is 1.16, It is based on the user selecting the best initial mask, which means that there is at least an additional interaction amount exceeding one point in the selection process. However, this interaction amount is not calculated in the SOTA table, so the SOTA data in the table is extremely unreasonable.
If the author can provide a reasonable explanation for this, I can consider changing my score.

Format problems:
1.Maintain adequate line length by filling each line with at least one-third of its content.
2.It's recommended to prioritize vector graphics over rasterized images for figures unless unavoidable.
3.unify the "first-click" and "first-click"

**Suitability:**

3

---

### Official Review · Reviewer_Bhdf · 2024-05-29

**Rating:** 3
**Confidence:** 4

**Summary:**

The manuscript presents DISNet, a pioneering interactive image segmentation framework that capitalizes on the inherent ambiguity of the initial user click to craft an array of initial mask proposals. These proposals are meticulously honed through a Dual-attentional Mask Correction module, complemented by an innovative sampling strategy to produce a spectrum of diverse ground truth masks. The framework's enhanced performance underscores the pivotal role of embracing the nuances of the first click. Nonetheless, the paper has certain ambiguities that necessitate a thoughtful and precise response from the authors.

**Strengths:**

1. It is valuable to recognize the ambiguity of the first click, and the analysis conducted in the paper is commendable.
2. The majority of the paper is clearly written and easy to understand.

**Limitations:**

1. The paper presents a significant flaw in its evaluation methodology: The automatic selection of the mask with the IoU from the proposals should be counted as an interaction because, in actual inference, it would require user operation and cannot rely on IoU computation. However, the authors have not clearly stated whether this selection is counted as an interaction in their reported performance metrics. This lack of clarity undermines the credibility of the results.
﻿
2. The first click is typically placed at the center of an object, which inherently minimizes part ambiguity. The advantage of this approach seems to be most effective when there are multiple candidate regions with centers that are in close proximity. Therefore, a more detailed comparative analysis with FCANet is warranted. The performance should be specifically examined in scenarios where the centers of the candidate regions are both close and distant to each other.

**Suitability:**

2

---

### Official Review · Reviewer_opwM · 2024-06-04

**Rating:** 4
**Confidence:** 1

**Summary:**

The authors present their "DISNet" (Diversified Interactive Segmentation Network), a framework for interactive segmentation that 1) tries to enhance the exploitation of the first click and 2) maximize the impact of subsequent clicks. In addition to this key contribution, the authors provide a sound experimental evaluation of their segmentation framework with six different datasets as well as a comparison with many different alternatives available in literature. They outperform most of them and their framework is very near to the few methods that do better of it in one of the datasets, in terms of number of clicks required to reach an intersection over union of 85% or 90%. The authors provide an ablation study as well.

**Strengths:**

1. The empirical evaluation is pretty sound: 2 different datasets for training, 4 for testing, comparison with many different algorithms available in literature.

2. The results show that the proposed framework outperforms/improves the results already available in literature, with few exceptions on specific cases, where the proposed framework obtains state-of-the-art performance anyway.

3. Sound description of the methodology

**Limitations:**

1. The main weakness in this work is the novelty. In fact, approaches that aim to reduce the clicks are available in the literature, even based on two stage architectures (in a similar fashion to what is proposed in this paper). For example:
 - T. Wang, H. Li, Y. Zheng and Q. Sun, "One-Click-Based Perception for Interactive Image Segmentation," in IEEE Transactions on Neural Networks and Learning Systems, doi: 10.1109/TNNLS.2023.3274127
 - J. Lin et al., "AdaptiveClick: Click-Aware Transformer With Adaptive Focal Loss for Interactive Image Segmentation," in IEEE Transactions on Neural Networks and Learning Systems, doi: 10.1109/TNNLS.2024.3378295
 - K. Li et al., "Multi-granularity Interaction Simulation for Unsupervised Interactive Segmentation," 2023 IEEE/CVF International Conference on Computer Vision (ICCV), Paris, France, 2023, pp. 666-676, doi: 10.1109/ICCV51070.2023.00068

Therefore, the authors should discuss these works and highlight the difference with their approach and compare against them as well.

2. I think that the paper is missing a discussion about the computation required by the authors' architecture with respect to those they compare against. At least, it might be in the form of a comment/speculation given that computing execution time might be unreliable given that the hardware used by the authors' might be different from the hardware used in other works..

Minor: in the ablation study the authors mention table 6 that is not in the paper.

**Suitability:**

2

---

### Meta-Review · Area_Chair_zRX2 · 2024-07-04

**Recommendation:** Accept (Poster)
**Confidence:** 3

**Metareview:**

The paper presents a promising approach to interactive image segmentation that demonstrates good performance in extensive evaluations. The most critical issue, highlighted by all reviewers, is the clarity of the evaluation methodology and the limited novelty. Reviewer uD9Y (and the others generally) pointed out a significant flaw in how the automatic selection of masks is handled, which could potentially undermine the credibility of the results. The authors' rebuttal included a detailed explanation of how this selection process is accounted for in the interaction metrics, ensuring that the evaluation accurately reflects real-world usage scenarios. This is very critical content, that must be updated in the manuscript. Nonetheless, reviewers praised the results, that the proposed framework improves the results already available in literature, with few exceptions on specific cases, where the proposed framework obtains state-of-the-art performance anyway. Considering this, and the general suggestion of acceptance from the reviewers, I recommend the paper for Acceptance. I strongly request that the authors update the paper to ensure the critical issue is corrected in the camera ready version of the paper.